# Colorectal Cancer Stem Cells Fuse with Monocytes to Form Tumour Hybrid Cells with the Ability to Migrate and Evade the Immune System

**DOI:** 10.3390/cancers14143445

**Published:** 2022-07-15

**Authors:** Karla Montalbán-Hernández, Ramón Cantero-Cid, José Carlos Casalvilla-Dueñas, José Avendaño-Ortiz, Elvira Marín, Roberto Lozano-Rodríguez, Verónica Terrón-Arcos, Marina Vicario-Bravo, Cristóbal Marcano, Jorge Saavedra-Ambrosy, Julia Prado-Montero, Jaime Valentín, Rebeca Pérez de Diego, Laura Córdoba, Elisa Pulido, Carlos del Fresno, Marta Dueñas, Eduardo López-Collazo

**Affiliations:** 1The Innate Immune Response Group, IdiPAZ, La Paz University Hospital, 28046 Madrid, Spain; karlamarina.hernandez@gmail.com (K.M.-H.); ramon.cantero@salud.madrid.org (R.C.-C.); jccasalvilla@yahoo.com (J.C.C.-D.); joseavenort@gmail.com (J.A.-O.); elviramarin@hotmail.com (E.M.); roberto.lozano.rodriguez@idipaz.es (R.L.-R.); vterronarcos@gmail.com (V.T.-A.); juliadelpradomontero@yahoo.es (J.P.-M.); jaimevquiroga@hotmail.com (J.V.); rebeca.perez@idipaz.es (R.P.d.D.); laura.cordobagarcia@gmail.com (L.C.); ely_vk@hotmail.com (E.P.); carlosdelfresnosanchez@gmail.com (C.d.F.); 2Tumour Immunology Lab, IdiPAZ, La Paz University Hospital, 28046 Madrid, Spain; 3Digestive Surgery Service, La Paz Univeristy Hospital, 28046 Madrid, Spain; marinavicariobravo@gmail.com (M.V.-B.); crismar3x8@gmail.com (C.M.); jorgev.saavedraa@gmail.com (J.S.-A.); 4Translational Research and Innovation in Surgery Group, La Paz Univeristy Hospital, 28046 Madrid, Spain; 5Biobank Platform, IdiPAZ, La Paz Universitary Hospital, 28046 Madrid, Spain; 6Molecular Oncology Unit, Biomedical Innovation Department, Centro de Investigaciones Energéticas, Medioambientales y Tecnológicas (CIEMAT), 28040 Madrid, Spain; marta.duenas@ciemat.es; 7Centre for Biomedical Research Network of Oncological Diseases (CIBERONC), 29029 Madrid, Spain; 8Centre for Biomedical Research Network of Respiratory Diseases (CIBERES), 28029 Madrid, Spain

**Keywords:** colorectal cancer, metastasis, fusion, monocytes, tumour hybrid cell, SIGLEC5

## Abstract

**Simple Summary:**

Colorectal cancer survival rates strongly decrease from initial to more advanced stages, primarily because of the occurrence of metastatic lesions. In this line, the search for clinical markers is of critical need. The aim of our study was to examine in vitro generated colorectal tumour hybrid cells (THCs) as a fusion between colorectal cancer (CRC) stem cells and human monocytes, as well as to evaluate their presence in tissue and blood samples from CRC patients. THCs, defined as CD45^+^CD14^+^EpCAM^+^, showed enhanced migratory, proliferative and immune evasion abilities compared to their parental cells. In a retrospective cohort of 23 patients, our data showed the potential relevance of resident tissue THCs in the generation of distant metastases. In addition, in a prospective cohort of 38 patients, our data confirmed the correlation between circulating THCs and sSIGLEC5 levels, a molecule which has already been previously described as a marker of poor prognosis in CRC patients. Altogether, our findings indicate that the number of THCs could serve as a novel biomarker for metastasis prediction in colorectal cancer patients.

**Abstract:**

Background: The cancer cell fusion theory could be one of the best explanations for the metastasis from primary tumours. Methods: Herein, we co-cultured colorectal cancer (CRC) stem cells with human monocytes and analysed the properties of the generated tumour hybrid cells (THCs). The presence of THCs in the bloodstream together with samples from primary and metastatic lesions and their clinical correlations were evaluated in CRC patients and were detected by both FACS and immunofluorescence methods. Additionally, the role of SIGLEC5 as an immune evasion molecule in colorectal cancer was evaluated. Results: Our data demonstrated the generation of THCs after the in vitro co-culture of CRC stem cells and monocytes. These cells, defined as CD45^+^CD14^+^EpCAM^+^, showed enhanced migratory and proliferative abilities. The THC-specific cell surface signature allows identification in matched primary tumour tissues and metastases as well as in the bloodstream from patients with CRC, thus functioning as a biomarker. Moreover, SIG-LEC5 expression on in vitro generated THCs has shown to be involved in the mechanism for immune evasion. Additionally, sSIGLEC5 levels correlated with THC numbers in the prospective cohort of patients. Conclusions: Our results indicate the generation of a hybrid entity after the in vitro co-culture between CRC stem cells and human monocytes. Moreover, THC numbers present in patients are related to both prognosis and the later spread of metastases in CRC patients.

## 1. Introduction

Colorectal cancer (CRC) is the third most common type of tumour worldwide, with a higher incidence of colon cancer compared to rectum cancer [1,2]. The incidence rate is steadily increasing in industrialised countries whilst the age of disease onset is decreasing. Although patients with CRC have a long-life expectancy, the 5-year survival rate becomes reduced from 90% in stages I/II to 12% in advanced stages III/IV, primarily due to the development of metastatic lesions in the later stages [3].

Surgery in CRC is curative at the early disease stages; however, mortality strongly increases in the advanced stages. In this regard, metastases account for around 90% of all cancer-related deaths, and to this day is still a poorly understood process in the pathophysiology of cancer. Within the events that comprise the metastatic cascade, the presence of circulating tumour cells (CTCs), which are shed from primary tumours and metastatic deposits into the bloodstream, provide a mechanism to further understand this heterogeneous process [4]. In this sense, CTCs are believed to be viable metastatic precursors. The United States Food and Drug Administration (FDA) approved the Cell Search^®^ system for immunoaffinity-based CTC detection in 2007. Since then, CTCs have been defined as 4′,6-diamidino-2-phenylindole (DAPI)+ nucleus cells which express surface cytokeratin (CK) 8/18 and/or 19 and EpCAM (epithelial cell marker), lacking CD45 expression (pan-leukocyte marker) [5].

Nonetheless, the co-expression of CD45 and other myeloid or stem cell markers have been found on circulating cancer cells, making the CTC definition incomplete [6,7]. In this regard, the cancer cell fusion theory, first described by the German pathologist Otto Aichel in the early 1900s [8], is now gaining strength as several groups postulate that leukocyte–tumour cell fusions have significant roles in tumour progression and cancer metastasis [9,10]. After that, Pawelek et al. demonstrated that hybrids produced in vitro and in vivo had higher metastatic potential compared to their parental tumour cells [11,12,13]. Recent studies have shown that these fused cancer cells are associated with more aggressive cancer phenotypes and might have a critical role in a patient’s outcome [14,15].

The Epithelial to Mesenchymal Transition (EMT) has also been pursued over many years as the best explanation for cancer metastasis. However, the mechanisms by which these master transcription factors become upregulated are still not understood. As an example, silencing of a certain EMT promoting transcription factors has been associated with a reduced number of metastases. The fact that some metastases still remained opens a door towards additional mechanisms which could better explain the metastatic cascade [16]. Considering all of this, the fusion between cancer cells and bone-marrow-derived cells, which are CD45+ cells, could certainly explain many of the unsolved questions.

Moving on, several studies have shown that tumour hybrid cells are mainly derived from the fusion between a tumour cell and a monocyte/macrophage [17,18]. There is a strong tumour associated macrophage (TAM) presence in the tumour microenvironment (TME), and these are known to play numerous roles in tumour initiation and progression. In this regard, a high TAM density in cancer is in most cases associated with poor prognosis, which could be related to cancer cells co-opting macrophage traits and therefore facilitating migration [19].

Herein we hypothesise the fusion between monocytes and tumour stem cells as an alternative mechanism for the generation of distant metastases in CRC patients, as previous data has shown that the stem cell phenotype favours this hybridisation process [17]. In this way, it would be possible for tumour cells to reach the bloodstream without any obstacles and to be insensitive to checkpoints. We have generated THCs in vitro identified, in distinction with CTCs, as CD45^+^CD14^+^EpCAM^+^. Additionally, these were observed to express and secrete high levels of SIGLEC5, being this receptor involved in their mechanism to escape immune surveillance. Eventually, THCs were also identified in the circulation of CRC patients and in paired tumour and peritumoural tissue sections. Both figuratively and literally, these are a type of “Trojan horse” that, hidden as an immune-cell-like cell, brings CRC cells to remote sites to colonise other organs, which previously led us to coin these tumour hybrid cells (THCs) as Trojan Horse Cells.

## 2. Materials and Methods

### 2.1. Patients and Healthy Volunteers Recruitment

Patients diagnosed with CRC were enrolled in this study prior to surgery at La Paz University Hospital (Madrid, Spain). CRC was diagnosed by a colonoscopy and tumour biopsy. Patients were recruited from 28 February 2017 to 7 February 2019, see details in Table 1 (retrospective cohort, n = 38). In addition, a validation cohort was recruited from 29 October 2021 to 14 March 2022, see details in and Table 2 (prospective cohort, n = 23). In this case, blood samples were taken 24 h before the patients underwent surgery; peripheral blood mononuclear cells (PBMCs) and plasma were isolated following standardised procedures [20]. As controls, healthy volunteers (n = 12) were recruited from the Blood Donor Services of La Paz University Hospital. All patients were classified according to their disease stage.

### 2.2. Cell Culture

A human SW620 cell line was purchased from the American Type Culture Collection (ATCC) and was cultured in 25 cm^2^ flasks with Dulbecco’s Modified Eagle’s Medium (DMEM, Gibco; Darmstadt, Germany) supplemented with 10% foetal bovine serum (FBS) (Gibco) and 0.01% penicillin/streptomycin (p/s) (Thermofisher; Waltham, MA, USA) at 37 °C in a humidified atmosphere with 5% CO_2_.

SW620 cancer stem cells (SW620CSCs) were obtained by culture in non-treated Costar plates with selective Dulbecco’s Modified Eagle’s Medium (DMEM)/F-12 (Gibco, 1:1) supplemented with 2 mmol/L of L-glutamine, 5 mmol/L of HEPES (both from Sigma; St. Louis, MO, USA), 20 ng/mL of epidermal growth factor (EGF, PeproTech; Rocky Hill, NJ, USA), N2 and B27 supplements (both from Gibco) and SB431542 (abcam) for 7 days, following previous protocols [17].

Healthy donors were recruited from the Blood Donor Service of La Paz University Hospital. Peripheral blood mononuclear cells were obtained by gradient centrifugation with Ficoll-Paque (GE Healthcare Bio-Sciences; Piscataway, NJ, USA). The monocytes were isolated from the PBMCs after one-hour adherence on Costar^®^ 24-well culture slides (Corning; Corning, NY, USA) on RPMI media (Gibco) supplemented with 0.01% penicillin/streptomycin (p/s) (Thermofisher).

SW620CSCs were co-cultured with human monocytes following a 1:10 ratio (CSC:Monocytes), following previous co-culture experimental conditions [17], on tissue culture-treated Costar^®^ 24-well plates (Corning) in RPMI (Gibco) supplemented with 10% FBS (Gibco) and 0.01% p/s (Thermofisher) at 37° in a humidified atmosphere with 5% CO_2_ for 5 days. After the co-culture time, the cells were recovered by gentle scraping and were analysed by flow cytometry or were used for functional experiments.

### 2.3. Reverse-Transcription qPCR

The SW620 and SW620^CSC^ cells were washed with PBS twice for RNA isolation. The High Pure RNA Isolation Kit (Roche Diagnostics; Basilea, Switzerland) was used. NanoDrop 2000 (Thermo Fisher Scientific; Waltham, MA, USA) was used to determine the RNA concentration of each sample. A total of 0.25 μg of RNA was used to synthesise the cDNA with the use of the High Capacity cDNA Reverse Transcription Kit (Applied Biosystems; Waltham, MA, USA).

The QuantiMix Easy SYGKit (Biotools; Madrid, Spain) was used for the RT-qPCRs according to the manufacturer’s instructions. Finally, the expression levels of each gene were analysed using the Biorad CFX96 touch deep well real-time PCR detection system (Bio-Rad). The expression levels of *beta-actin* housekeeping were used as the internal standard to normalise the data. The relative expression was determined using the 2(-Delta Delta Ct) method. All the specific primers were synthesised by Eurofins Genomics. The specific primers for each gene are shown in Appendix A.

### 2.4. Vital Colorant Assay

The monocytes were stained with DID vital colorant and SW620^CSCs^ were stained with DIO vital colorant following the manufacturer’s instructions (Vybrant Multicolor Cell-Labelling Kit-Thermofisher). The DID-stained monocytes were co-cultured with DIO-stained SW620^CSCs^ following a 1:10 ratio (SW620^CSCs^:Monocytes) on Falcon^®^ 8-well culture slides (Corning).

DID-Monocytes/DIO-SW620^CSCs^ co-cultures were fixed with 4% paraformaldehyde (PFA) (SIGMA-P6148) and were mounted with the Antifade Vectashield^®^ mounting medium (Vector) containing 4′,6-Diamidino-2-Phenylindole, Dihydrochloride (DAPI) for nuclear staining. Images were taken with a Leica–DMI400D confocal laser-scanning microscope at different magnifications. The excitation/emission wavelengths were 405/461, 644/665 and 484/501 nm for DAPI, DID and DIO, respectively. All fluorescent image analyses were performed with LAS AF (Leica; Wetzlar, Germany).

### 2.5. Cell Sorting and Flow Cytometry

For cell sorting, the co-cultured cells were stained in PBS (Gibco-BRL Life Technologies) with EpCAM-Phycoerythrin (PE) (Biolegend; San Diego, CA, USA) and CD14-Fluorescein-isothiocyanate (FITC) (Immunostep) for 30 min in the dark at 4 °C at the recommended manufacturer’s concentration for each antibody. Afterwards, the cells were washed with Phosphate Buffer Saline (PBS) and were resuspended in RPMI supplemented with 10% FBS and 0.01% p/s media for cell sorting in a BD FACS Influx TM Cell sorter (BD Biosciences; Franklin Lakes, NJ, USA).

Fluorescence Activated Cell Sorting (FACS) analyses were developed using specific human antibodies (Abs) to the following surface molecules: EpCAM-Brilliant Violet (BV) 605, CD14-Brilliant Ultra Violet (BUV) 395, CD45-PECF594, CD19-BV786, CD16-BV480, CD56-BUV737, PD-L1-BV421 (all from BD), CD3-Peridinin Chlorophyll Protein (PercP) (Immunostep), CD1c-Super Bright (SB) 436, CD45-Pacific Orange (PO), CD163-Efluor450, Live Dead Blue, CFSE (all from invitrogen), HLA-DR-PECy5, Siglec5-APC (both from Biolegend) and TGFβ-PE (Miltenyi). In the case of pan cytokeratin (PanK)-PE (Miltenyi), where the staining is intracellular, the Cytofix/Cytoperm Solution Kit (BD) was used following the manufacturer’s instructions. For the different assays, the samples were run in either FACS Celesta (BD Biosciences) or FACS Aurora (Cytek) flow cytometers, and the data were analysed using FlowJo (TreeStar; Woodburn, OR, USA) v.10.7.2 software.

### 2.6. In Vitro Modulation of Monocyte Polarisation

Isolated monocytes (Miltenyi Biotec Monocyte Isolation Kit II) were polarised towards the M1 phenotype with IFN-γ (100 µg/mL) and towards M2 with IL-4 (20 µg/mL) 48 h prior to co-culture with SW620^CSCs^.

### 2.7. Soluble Cytokines Quantification

The supernatants were collected after 24 h of monocyte polarisation and the cytokine levels were measured using the LegendPlex Human Essential Immune Response Panel Kit (IFNγ, IL-8, IL-4, IL-10 and TGFβ) according to the manufacturer’s instructions (Biolegend). The samples were acquired using a FACS Calibur flow cytometer (BD Biosciences). The data were analysed using Biolegend v8.0 software (Biolegend).

### 2.8. Transwell Migration Assay

Co-cultures, monocytes and SW620^CSCs^ were scraped after 5 days of culture and the cells were collected into RPMI (w/o FBS) and were seeded (10^5^ cells) on the top compartment of Corning transwell chambers (8 µm, Sigma). The bottom compartment was filled with 700 µL of RPMI + 10% FBS. After 48 h, the contents from the top and bottom were recovered and analysed through flow cytometry using 1000 inert beads (Reagent D from Mice CBA, BD Biosciences) as the control to stop acquisition. The migration (M) percentage was assessed as the count of each cell type in the bottom (B) with respect to the remaining amount (A + B) as shown in: *M* = [*B*/(*A* + *B)*] × 100.

### 2.9. In Vitro Proliferation Assays

Human monocytes (8.1 × 10^6^) and SW620^CSCs^ (0.9 × 10^6^) were co-cultured for 5 days and were sorted. Afterwards, 2500 sorted EpCAM^+^ SW620-CSCs, CD14^+^ monocytes and EpCAM^+^CD14^+^ hybrid cells were co-cultured with 25,000 carboxyfluorescein succinimidyl ester (CFSE) -labelled PBMCs from healthy volunteers (1:10 ratio). After 5 days, T cell proliferation induced by pokeweed mitogen (2.5 µg/mL) was analysed by FACS in the presence/absence of 10 µg/mL of α-PD-1 (Bristol-Myers Squibb; New York, NY, USA), 1 µg/mL of α-SIGLEC5 (ThermoFisher PA5-47058) and 10 µg/mL of α-CD25 (Novartis; Basel, Switzerland) human antibodies. SB431542 was used as a TGFβ-LAP inhibitor (Merck; Darmstadt, Germany) at a concentration of 10 µM.

FACS analyses were developed using specific human antibodies (Abs) to the following surface molecules: CD4–PercP and CD8-APC (both from immunostep). The samples were run in the FACS Calibur (BD Biosciences) flow cytometer and the data were analysed using FlowJo (TreeStar) v.10.7.2 software.

### 2.10. Soluble Immune-Checkpoint Measurement

Soluble ICs (sTGFβ, sPD-L1, sSIGLEC5, sCD25, Galetin9 and sTim3) were measured in supernatants from sorted cells with the use of the LegendPlex Custom Human Immune Checkpoint Panel, following the manufacturer’s instructions (Biolegend). To do this, the supernatants obtained from the sorted cell populations were incubated with premixed antibody-coated beads. Following this, the supernatants were incubated with detection antibodies and streptavidin with phycoerythrin conjugate. Finally, the samples were acquired using a FACS Calibur flow cytometer (BD) and the Biolegend v8.0 software (Biolegend) was used for the analyses.

### 2.11. Tissue Sample Immunohistochemistry

Paraformaldehyde 4% (SIGMA-P6148) was used to fix the tissue samples. Following this, the fixed tissue samples were embedded in OCT and were cut with a Leica RM2255 microtome (Leica Biosystems; Wetzlar, Germany) into 10 um sections. Confocal scanning laser microscopy was used to analyse the antibody-stained sections with the use of a Leica SPE laser-scanning microscope (Leica Biosystems). Finally, the images were quantified with ImageJ.

Colorectal cancer tissue microarrays (TMAs) were generated with the obtained tissue samples. These arrays were stained for CD45, CD14 and EpCAM (all from Abcam). A mounting medium containing DAPI and α-fading (Vectashield; Vector Laboratories; Newark, CA, USA) was added to the sections.

### 2.12. Statistical Analysis

Parametric or non-parametric statistical analyses were used to analyse continuous variables after examining their Gaussian distribution by the Shapiro–Wilk test. Chi-squared testing was used to analyse the categorical parameters. Consequently, either a t-student or an ANOVA followed by a Tukey analysis or Kruskal–Wallis test were used. Survival analyses were performed using Kaplan–Meier survival curves and receiver operating characteristic (ROC) analyses (GraphPad Prism 9). The Gehan–Breslow–Wilcoxon and Cox–Mantel log-rank test models were used for survival prediction and the Wilson/Brown test for ROC curve analyses. Additionally, GraphPad Prism 9 and Microsoft Office Excel were used to calculate the Youden index. The area under the curve (AUC) and 95% confidence intervals (CIs) are reported. Significance was set (* *p* < 0.05, ** *p* < 0.01, *** *p* < 0.001, **** *p* < 0.0001) using Graphpad Prism 9.0 software.

### 2.13. Ethics Approval

All volunteers signed informed consent forms and the data were treated according to the recommended criteria of confidentiality, following the ethical guidelines of the 1975 Declaration of Helsinki. The study was approved by the Hospital Universitario La Paz (HULP) Ethical Committee.

## 3. Results

### 3.1. Co-Culture of Colorectal Cancer Stem Cells with Human Monocytes in Vitro Yields a New Hybrid Entity

In order to test the potential fusion between CRC stem cells and human monocytes, the CRC SW620 cell line was first dedifferentiated into a cancer stem cell (CSC) under non-adherent conditions for 7 days and was then co-cultured with human monocytes for another 5 days (Figure 1A). Figure 1B illustrates the morphological differences between the use of a standard or CSC medium, focused on the formation of spheroid aggregates. To ensure CSC dedifferentiation, the relative expression of the Yamanaka’s pluripotency cassette was tested, with significantly higher levels for *NANOG* and elevated levels of *OCT3/4* and *KLF4* in the SW620^CSC^ cells compared to the SW620 control (Figure 1C). In addition, CXCR4 expression found a 1.91-fold change from the control to CSC phenotype, which is strongly associated with the stem cell phenotype in CRC according to Delle Cave et al., 2021 [21] (data not shown).

Next, the CRC stem cells (SW620^CSC^) and human monocytes were stained with the vital colorants DIO and DID, respectively. These cells were co-cultured for 5 days, following previous-established experimental conditions [17]. Confocal microscopy confirmed the generation of double positive entities that we defined as tumour hybrid cells (THCs, DID^+^DIO^+^) (Figure 1D). Similar results were obtained by spectral flow cytometry; after a 5-day co-culture between EpCAM^+^ SW620^CSCs^ and CD14^+^ human monocytes, a double positive population (EpCAM^+^CD14^+^) was detected (Figure 1E). The gating strategy that followed is shown in Appendix A.

Note that, after co-culture, three defined populations were detected: CD14^+^ (monocytes), EpCAM^+^ (CRC stem cells) and CD14^+^EpCAM^+^ (THCs) (Figure 1E and Figure 2A,B). We remark that CD14^+^EpCAM^+^ also expressed CD45 (Figure 2C,D). Figure 2E–G show levels of these markers on both monocytes and SW620^CSC^ before co-culture, confirming how CD14 and CD45 are exclusive leukocyte markers, not expressed on tumour cells, and EpCAM is an epithelial marker not found on leukocytes. Thus, we define THCs as CD45^+^CD14^+^EpCAM^+^. As we have described before [17], the CSC phenotype significantly generated higher levels of THCs compared to SW620 control cells (Figure 2H). However, in contrast to our previous report in lung cancer [17], we have identified EPCAM instead of pan cytokeratin (PanK) as the optimal biomarker for SW620^CSCs^. Both the SW620 cell line and SW620^CSC^ expressed very low levels of PanK on their cell surface (Appendix A).

### 3.2. Anti-Inflammatory Phenotype Favours THC Generation

Once THC (CD45^+^CD14^+^EpCAM^+^) generation as a result of SW620^CSC^ co-culture with human monocytes was established, we continued to study the influence of the monocyte polarisation status on this fusion event. Then, purified human monocytes, obtained by negative selection columns, were polarised towards the M1 and M2 phenotypes using a stimulation with IFNγ and IL-4 for 24 h, respectively (Figure 3A). The polarisation status was verified by cell surface marker expression. On one hand, the CD163 overexpression was confirmed on the M2 monocytes and HLA-DR on the M1 monocytes (Figure 3B). On the other hand, polarisation was also checked by cytokine quantification. Note that monocytes were polarised for 24 h, washed with PBS and left for another 24 h at rest before supernatant collection. The levels of IFNγ and IL-8 were found to be significantly higher in the M1 monocytes supernatant (Figure 3C,D). In contrast IL-4, IL-10 and TGFβ were significantly higher in the M2 monocytes cultures (Figure 3E–G). Eventually, co-culture of SW620^CSCs^ with M2-polarised monocytes yielded a higher THC percentage compared to M1 (Figure 3H).

### 3.3. Colorectal THCs Exhibit a High Rate of Migration and Proliferation In Vitro

To study whether the fusion conferred any advantage to the generated CRC THCs in terms of migration, a transwell migration assay was performed on the sorted populations, as Figure 4A shows. In this sense, THCs (CD45^+^CD14^+^EPCAM^+^) exhibited a higher migratory capacity than their parental SW620CSC (EPCAM+) (Figure 4B). Along these lines, proliferative ability as a major cancer hallmark was also tested. Again, the sorted populations were individually labelled with CFSE and were left to proliferate for 5 days. THCs showed a significantly higher proliferation rate than their parental SW620CSC cells (Figure 4C).

### 3.4. Colorectal THCs Inhibit Lymphocyte Proliferation through SIGLEC5 Immune Checkpoint

We and others have postulated that THCs are responsible for metastasis in several types of solid tumours [17,18]; thus, THCs should prevent the immune response. To evaluate this, both the expression and secretion of a set of immune checkpoints (ICs), IC ligands and immune modulators on sorted THCs, monocytes and SW620CSC after co-culture were studied. Figure 5A (left panel) shows that SIGLEC5 was significantly incremented on monocytes and THCs, as well as its soluble form (Figure 5A, right panel). The immune modulator TGFβ exhibited a tendency toward overexpression on both monocytes and THCs (Figure 5B, left panel), with its soluble form being significantly higher in the monocyte and THC culture supernatants (Figure 5B, right panel). When PD-L1 was evaluated, we found high expression on the membrane of monocytes (Figure 5C, left panel), and its soluble form was also found highest in THC culture supernatant (Figure 5C, right panel). However, no significant difference was shown. Finally, when the soluble levels of sCD25, Galectin 9 and sTIM3 were analysed, only sCD25 levels were significantly higher in THCs (Figure 5D).

Next, we explored the relevance of these molecules in the interaction between THCs and the immune system. Both human CD4^+^ and CD8^+^ T cells showed a reduction in a mitogen-induced proliferation after exposure to THCs with regard to SW620CSCs (Figure 6A). Blocking antibodies against the main molecules studied and presented in Figure 5 were used to evaluate whether these were involved in immune escape. Thus, an anti-SIGLEC5 antibody rescued CD4 T cell proliferation in the presence of CRC THC (Figure 6B). However, Figure 6C,D show how the use of a TGFβ inhibitor or anti-PD-L1 antibodies did not restore CD4 nor CD8 proliferation. Note that SIGLEC5 did not play a significant role in the fusion event opposite to in vitro generated THCs immune escape (Appendix A).

### 3.5. THC-Specific Signature Is Present in Primary Tissue and Metastatic Samples of Colorectal Cancer Patients

After establishing the THC-specific signature CD45+CD14+EpCAM+ (Figure 2D), we explored the presence of these entities in patients. Thus, a retrospective cohort (Table 1; n = 23) with patients recruited from 28 February 2017 to 7 February 2019 by the Digestive Surgery Service of La Paz University Hospital was studied. The presence of THCs (CD45^+^CD14^+^EpCAM^+^) was observed in paired tumour, peritumour and metastasis tissue samples from CRC patients (Figure 7). Moreover, THC counts in both tumour (left) and peritumour (right) tissue samples was significantly greater in those patients who later developed metastasis (Figure 8A). Noteworthy, the THC counts were double blinded and a co-expression of DAPI, CD45, CD14 and EpCAM had to occur in all fields of the same picture for each count. Given that the THC counts were higher in patients who suffered metastasis, we explored whether the THC counts could serve as metastasis predictors in CRC. To discriminate between patients who developed metastasis and those who did not, a ROC analysis of THC counts was performed for both the tumour (Figure 8B, AUC = 0.879; 95% CI 0.720–1.00; *p* = 0.039) and peritumour tissues (Figure 8C, AUC = 0.850, 95% CI = 0.592–1.00; *p* = 0.055). The optimal cut-off values, estimated by the Youden index, were 28.75, with a high sensitivity (1.00; 95% CI 0.438–1.000) and specificity (0.77; 95% CI 0.547–0.910), and 48 counts, with a high sensitivity (0.66; 95% CI 0.118–0.982) and specificity (1.00; 95% CI 0.838–1.000) for tumour and peritumour tissue samples, respectively. The CRC patients were then classified into high THC count and low THC count groups according to their Youden cut-off value. Contingency table analyses showed a significantly higher metastasis probability in the high THC count group for both tumour and peritumour tissue samples compared to the low THC count group (Figure 8D,E, X^2^ = 6.07 and 14.6, respectively, *p* = 0.013 and 0.0001, respectively). Finally, Kaplan–Meier analyses showed that patients classified as having a high THC count for both tumour and peritumour samples had a significantly shorter metastasis-free survival than those in the low THC count group, indicating the potential use of pre-operative THC count as a metastasis predictor in CRC patients (Figure 8F,G). The two statistical analyses performed, the Gehan–Breslow–Wilcoxon and the log-rank (Mantel–Cox), were statistically significant. Therefore, the THC counts in tumour and peritumour tissue samples served as metastasis predictors in the CRC patients.

### 3.6. THC-Specific Signature Is Present in Circulation of Colorectal Cancer Patients

We moved on to identify these cells in circulation. A prospective cohort (n = 38) was recruited from 29 October 2021 to 14 March 2022 by the Digestive Surgery Service of La Paz University Hospital (Table 2). The presence of THCs was significantly greater in the blood from the CRC patients compared to the HVs (Figure 9A). The gating strategy that followed for THCs determination is shown in Appendix A. Because of the highly positive prognosis that CRC patients have after surgery and the little follow-up time these patients have, no metachronous metastases had yet occurred (Table 2).

Finally, considering the previously addressed relationship between immune evasion properties in THCs and SIGLEC5, we moved on to analyse the correlation between soluble SIGLEC5 (sSIGLEC5) levels and THC counts in circulation. Figure 9B shows a positive correlation between THC counts and sSIGLEC5 levels.

This prospective patient cohort had a short follow-up time that did not allow to monitor enough metastatic events. Nonetheless, since sSIGLEC5 levels have been recently associated with disease prognosis in CRC [22], the positive correlation found between THC counts and sSIGLEC5 levels indicates a potential correlation between THC numbers in circulation and patient prognosis in CRC.

## 4. Discussion

To this day, still little is known concerning the onset of metastasis. In this regard, the cancer cell fusion theory is now being referred to by many as a mechanism which could explain most aspects of malignancy [13,23]. It accounts for the majority of cancer-associated death and continues to be the least understood phase of the disease [23]. Nonetheless, this mechanism has only been circumstantially studied [12,23]. Our research provides evidence regarding the existence of a hybrid entity not only in our previously published lung cancer data [17], but also in CRC patients.

Tumour hybrid cells (THCs) were first associated with metastasis after human astrocytic glioma cells were implanted in a hamster-cheek pouch, which led to the generation of metastatic cells which contained mixed human–hamster karyotypes [24]. Currently, THCs have been described in lung, bladder, breast, ovarian and pancreatic carcinomas, principally as the fusion between a tumour cell and a macrophage. In this sense, monocytes/macrophages have been shown to display the highest fusion rate within the different leukocyte populations [15,23]. In addition, these cells have also been associated by our group and concomitantly by other authors as important partners in the tumour–cell fusion process [25,26]. These THCs are characterised in circulation through the co-expression of the leukocyte marker CD45 and the epithelial markers EpCAM or Cytokeratins [15,17,18,27,28]. Not only that, THCs have been highly associated with enhanced migration, proliferation, immune evasion and metastatic potential in patients [17,29,30]. Even though the fusion theory is still controversial, we must bear in mind that monocytes/macrophages could be exploiting this mechanism with tumour cells, favouring metastatic processes [31].

Herein, we report the existence of a hybrid entity with enhanced mobility, proliferation and immune evasion hallmarks as a result of CRC stem cells and human monocytes co-culture, which was also identified in both the tissues and circulation of CRC patients. In this sense, the THC signature in CRC was defined as CD45^+^CD14^+^EpCAM^+^. The co-expression of EpCAM^+^CD45^+^ hybrid cells has already been described in the ascites of ovarian cancer patients, which the authors associate with a maintenance of epithelial features and the gain of pro-migratory characteristics [27].

Even though the scientific community is still reluctant towards this theory as a novel way of explaining metastasis, cancer cell fusion is now being studied by many. Gast et al. (2018) have demonstrated the spontaneous fusion between murine macrophages and MC38 mouse colon carcinoma cells, which generated highly motile hybrids with enhanced metastatic abilities [32]. In a similar manner, Shabo et al. (2013) identified the co-expression of the macrophage antigen CD163 in human breast and colorectal cancer tissues, which were also associated with poor survival and liver metastases [33]. Not only has their presence been previously studied in solid tumours, but also in circulation. Clawson and colleagues (2012) have identified CK+CD45+ tumour hybrid cells in the peripheral blood of patients with melanoma [14].

Due to the important role of the immune system in the elimination of aberrant cells, evading the immune system is critical for cancer and metastasis. In this regard, THCs did not promote either CD4 or CD8 proliferation. In addition, its presence reduced CD4 and CD8 T cell proliferation after mitogen stimulation. Along these lines, the THCs obtained expressed high levels of immune modulator molecules, particularly SIGLEC5 and sSIGLEC5. Moreover, CD4 proliferation after the mitogen challenge was restored in the presence of an anti-SIGLEC5 antibody. It is worth mentioning that the recently reported immune checkpoint ligand, SIGLEC5 [34], was expressed both anchored to the in vitro generated THCs membrane and in its soluble form. Previously, we have described high levels of sSIGLEC5 in plasma from patients with CRC compared with healthy volunteers. Additionally, sSIGLEC5 levels were higher in exitus than in survivors, and the receiver operating characteristic curve analysis revealed sSIGLEC5 to be an exitus predictor in CRC patients [35]. These results emphasise the relevance of searching novel markers and also highlights that THCs possess their own immune evasion mechanisms. Considering SIGLEC5 is predominantly expressed on innate immune cells [36], it is consistent that, following fusion with monocytes, this molecule would be expressed/transferred on the surface of THCs, gifting these hybrids with the ability of circulating in the bloodstream of patients unnoticed.

In contrast to our THCs (CD45^+^CD14^+^EpCAM^+^), CTCs are defined as EpCAM^+^CD45^−^ [37]. The total number of CTCs equal or higher to 5 per 7.5mL of the whole blood previous to any treatment have been associated with poor prognosis and shorter median-progression survival in metastatic breast and colon cancer [38,39]. Herein, we have reported elevated THC counts in CRC patients compared to healthy volunteers in a prospective cohort. Because the cohort was recruited only a year ago, there are no conclusive data on the correlation between the presence of circulating THCs and the occurrence of metastases. Nonetheless, THC counts found in circulation of CRC patients positively correlated with sSIGLEC5 levels. The significant correlation found between THC count and sSIGLEC5 levels could indicate the association of circulating THCs with patient prognosis over time. Not only that, histological analyses of THC count in paired tumour and peritumour tissue samples from CRC patients indicated a significant association between high THC counts and the later development of distant metastases in our retrospective CRC patient cohort.

Altogether, our data propose a novel hybrid population of circulating tumour cells, THCs, which, in a similar way to CTCs, are a rare population found in the circulation of colon cancer patients and could be responsible for metastatic spread. Considering CTC detection ignores all subsets of cells expressing CD45, THCs could be considered as ignored hidden enemies in CRC patients. These findings reinforce the extremely complex concept of the metastatic pith and shed light onto possible improvements in clinical practice to fight metastasis.

## 5. Conclusions

Our data shows the generation of tumour hybrid cells (THCs) in vitro after co-culture between human monocytes and SW620CSCs. These can also be identified in circulation and paired tumour and peritumour tissue samples with the co-expression of CD45, CD14 and EpCAM markers. Moreover, these THCs exhibit enhanced mobility and proliferative abilities compared to their parental cells. Additionally, the expression of SIGLEC5 on THCs seems to be involved in promoting their immune evasion, considering the use of a blocking antibody against SIGLEC5 recuperated CD4 T cell proliferation in the presence of THCs. In our retrospective cohort, THC counts were strongly associated with the later development of distant metastases in our CRC patients. Additionally, circulating THCs correlated with sSIGLEC5 levels in our prospective cohort; the latter which has been recently related to poor prognosis in CRC patients. Altogether, our data propose an alternative explanation for metastasis, highlighting the importance of studying novel signatures and mechanisms which could aid in the fight towards understanding metastasis.

## Figures and Tables

**Figure 1 cancers-14-03445-f001:**
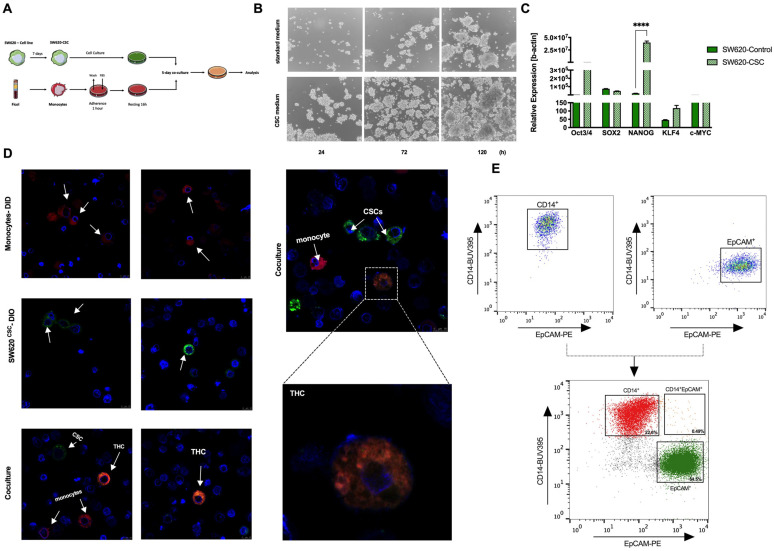
CRC stem cells fuse with human monocytes in vitro. (**A**) Schematic representation of experimental co-culture conditions. (**B**) Brightfield microscopy images at 24, 72 and 120 h showing morphological differences between standard (upper panel) and CSC (lower panel) medium. (**C**) Relative expression of OCT3/4, SOX2, NANOG, KLF4 and cMYC pluripotency genes in SW620-control (filled green) and SW620-CSC (dotted green) cells (n = 3). (**D**) Representative confocal images of co-cultured DIO stained SW620CSC (green) with DID stained monocytes (red) and DAPI for nuclei (blue) is shown (n = 3). Zoomed image of THC (tangerine) can be observed (63× magnification). (**E**) Representative FACS analysis gating strategy illustrating EpCAM+ and CD14+ single stainings and after 5 days of co-culture illustrating EpCAM^+^ SW620CSC (green), CD14^+^ human monocytes (red) and CD14^+^EpCAM^+^ (THCs, tangerine); ****, *p* < 0.0001; see details in Appendix A.

**Figure 2 cancers-14-03445-f002:**
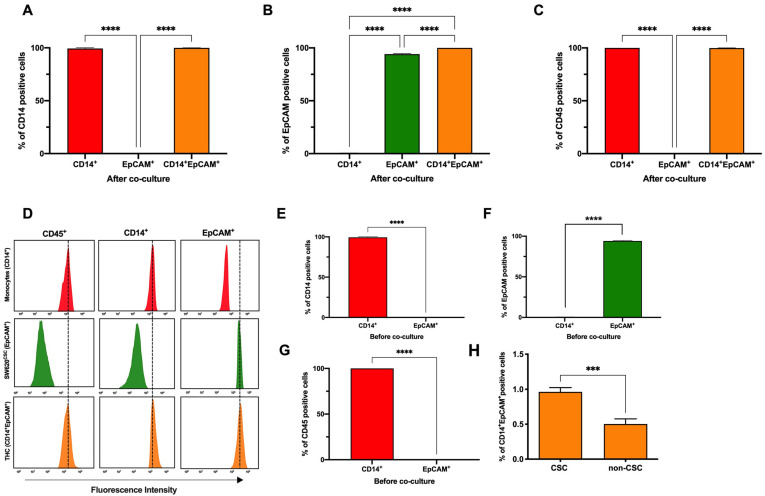
Characterisation of novel resulted THCs after co-culture. Analysis of expression of (**A**) CD14, (**B**) EpCAM and (**C**) CD45 on CD14^+^ human monocytes (red), EpCAM^+^ SW620^CSC^ (green) and on CD14^+^EpCAM^+^ THCs (tangerine) after 5-day co-culture by FACS (n = 3). ****, *p* < 0.0001 in one-way ANOVA with Tukey’s multiple comparison post-hoc test. (**D**) Representative histogram of CD45, CD14 and EpCAM fluorescence intensity in monocytes (red), SW620^CSC^ (green) and THCs (tangerine) after 5-day co-culture. (**E**) Analysis of CD14 expression on CD14^+^ human monocytes (red) and EpCAM^+^ SW620^CSC^ (green) before co-culture by FACS. (**F**) Analysis of EpCAM expression on CD14^+^ human monocytes (red) and EpCAM^+^ SW620^CSC^ (green) before co-culture by FACS. (**G**) Analysis of CD45 expression on CD14^+^ human monocytes (red) and EpCAM^+^ SW620^CSC^ (green) before co-culture by FACS. (**H**) Percentage of CD14^+^EpCAM^+^-positive cells after 5-day co-culture with SW620^CSC^ and SW620 cells. n = 3; ***, *p* < 0.001; and ****, *p* < 0.0001 in unpaired *t*-test.

**Figure 3 cancers-14-03445-f003:**
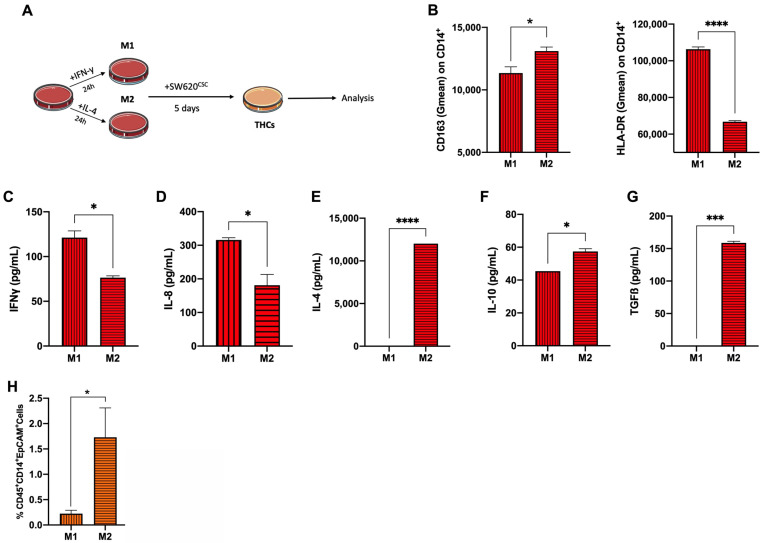
Human monocyte polarisation towards M2 yields higher percentage of THCs. (**A**) Schematic representation of experimental monocyte polarisation and co-culture conditions. (**B**) Geometric mean (Gmean) of surface markers CD163 (left) and HLA-DR (right) on M1 (vertical lines) and M2 (horizontal lines) polarised monocytes. Levels of (**C**) IFN_γ_; (**D**) IL-8; (**E**) IL-4; (**F**) IL-10; and (**G**) TGFβ in cell supernatants from M1 (vertical lines) and M2 (horizontal lines) polarised monocytes after 24 h of stimulation, wash and 24 additional hours of resting. (**H**) Percentage of CD45^+^CD14^+^EpCAM^+^ cells of SW620CSC cocultures with M1 (vertical lines) and M2 (horizontal lines) polarised monocytes. n = 3; *, *p* < 0.05; ***, *p* < 0.001; and ****, *p* < 0.0001 in unpaired *t*-test.

**Figure 4 cancers-14-03445-f004:**
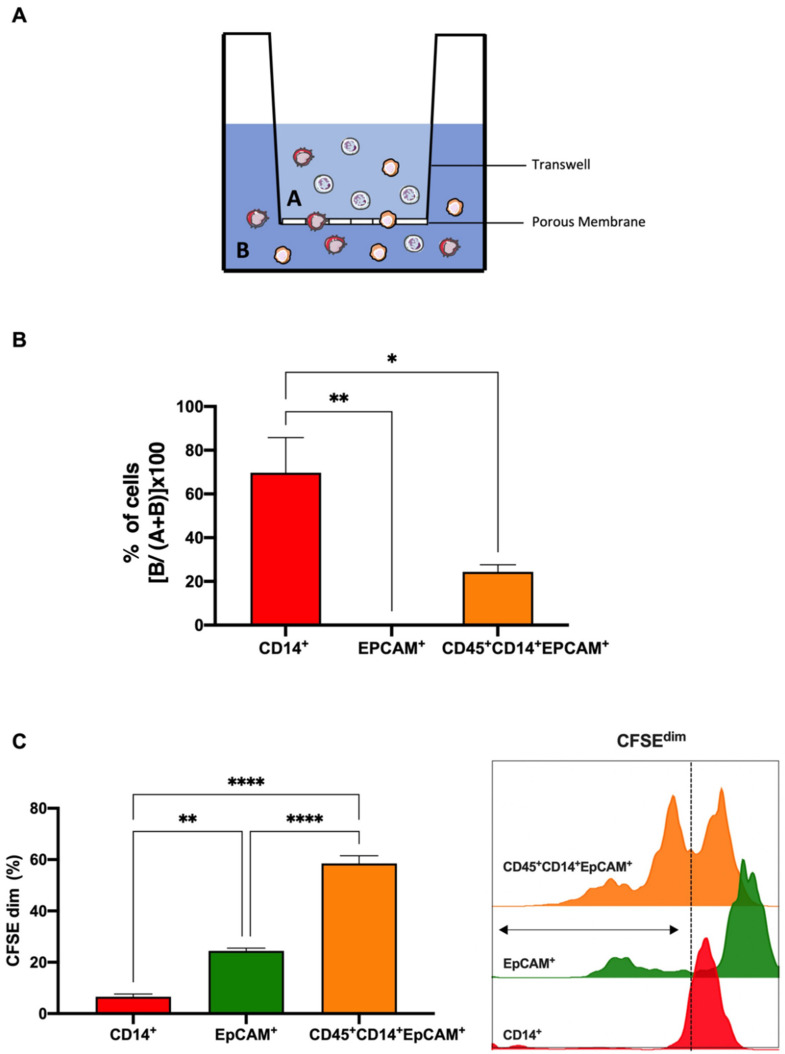
THCs display enhanced migratory and proliferative abilities. (**A**) Schematic representation of the migration assay (8 µm transwell pores). Cells resulting from the fusion assay were put on the top chamber (**A**) then migrating cells were quantified in the bottom chamber (**B**) by FACS. (**B**) Percentages of migration, % of cells [*B*/(*A + B*)] × 100, of the CD14+ human monocytes (red), EpCAM+ SW620CSCs (green) and CD45^+^CD14^+^EpCAM^+^ THCs (tangerine) from the transwell assay are shown (n = 4). (**C**) Left: percentage of proliferation measured as CFSE dilution (CFSEdim) of CD14^+^ human monocytes (red), EpCAM^+^ SW620CSCs (green) and CD14^+^EpCAM^+^ THCs after 5 days (n = 4). Right: representative histograms of CD14^+^ human monocytes (red), EpCAM^+^ SW620CSCs (green) and CD45^+^CD14^+^EpCAM^+^ THCs proliferation after 5 days. *, *p* < 0.05; **, *p* < 0.01; and ****, *p* < 0.0001 in ordinary one-way ANOVA with Tukey’s multiple comparison post-hoc test.

**Figure 5 cancers-14-03445-f005:**
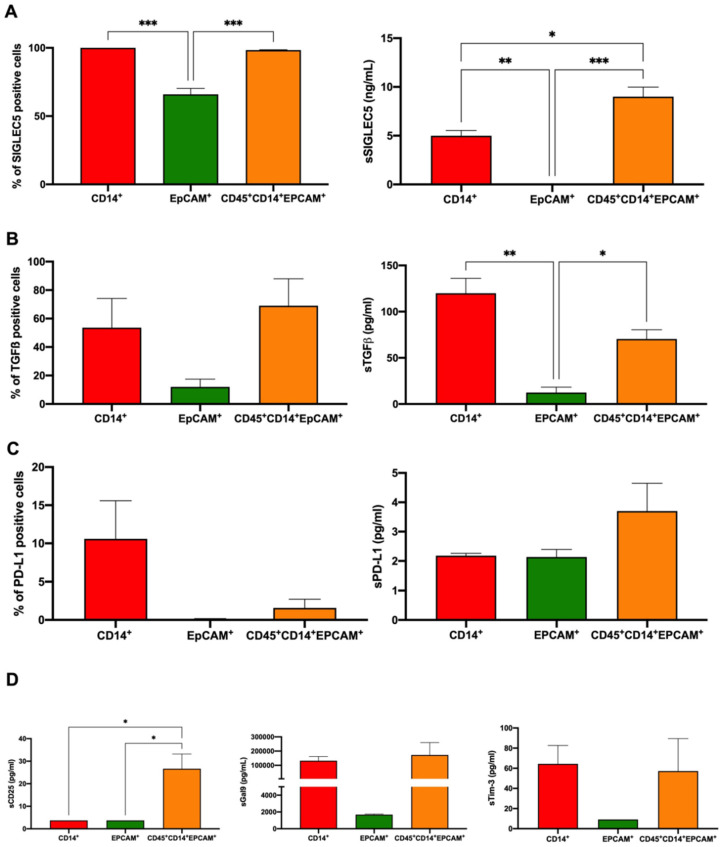
Characterisation of IC expression on THCs after co-culture. Left panels: expression of (**A**) SIGLEC5; (**B**) TGFβ; and (**C**) PD-L1 on the surface of human monocytes (red), SW620^CSC^ (green) and THCs (tangerine). Right panels: Soluble levels of A, sSIGLEC5 (ng/mL); B, sTGFβ (pg/mL); and C, sPD-L1 (pg/mL) in the supernatant of sorted human monocytes (red), SW620^CSC^ (green) and THCs (tangerine) after a 5-day culture. (**D**) Soluble levels of sCD25 (pg/mL), sGal9 (pg/mL) and sTim3 (pg/mL) in the supernatant of sorted human monocytes (red), SW620^CSC^ (green) and THCs (tangerine) after co-culture. n = 3; *, *p* < 0.05; **, *p* < 0.01; and ***, *p* < 0.001.

**Figure 6 cancers-14-03445-f006:**
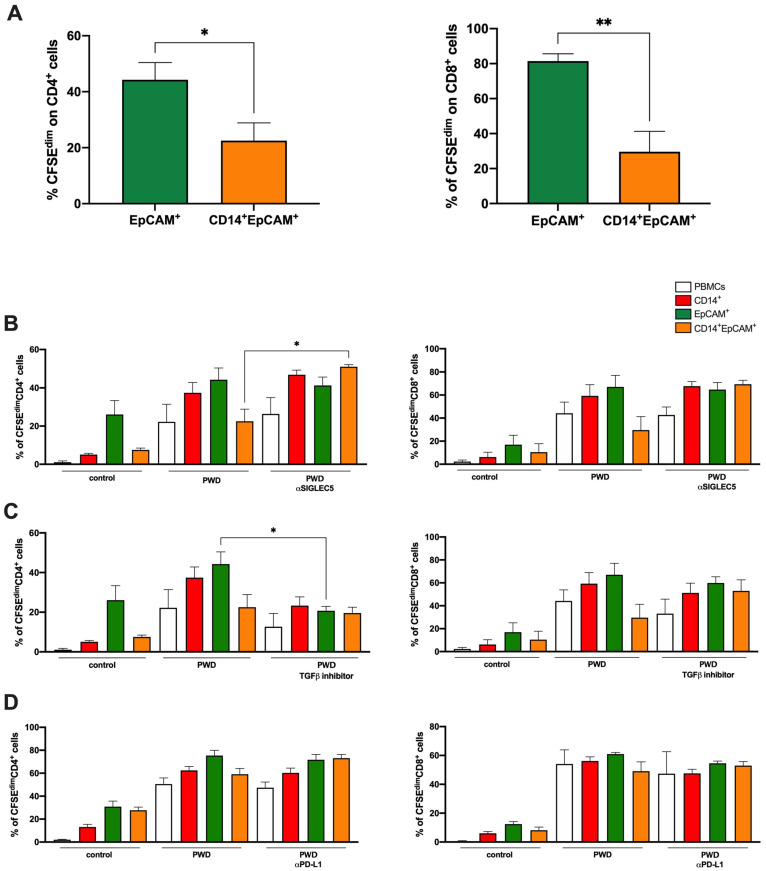
Sorted THCs inhibit CD4^+^ T cell proliferation in a SIGLEC5-dependent manner. (**A**) Proliferation of CD4 (left) and CD8 (right) T lymphocytes after a 5-day co-culture with pokeweed (PWD) stimulated sorted SW620^CSC^ (EpCAM^+^, green) and THCs (CD45^+^CD14^+^EpCAM^+^, tangerine). Proliferation of CD4 (left panels) and CD8 (right panels) T lymphocytes after a 5-day co-culture with pokeweed (PWD)-stimulated PBMCs (white), sorted human monocytes (CD14^+^, red), SW620^CSC^ (EpCAM^+^, green) and THCs (CD45^+^CD14^+^EpCAM^+^, tangerine) in the presence or absence of a blocking antibody or inhibitor against (**B**) SIGLEC5; (**C**) TGFβ; and (**D**) PD-L1. N = 6; *, *p* < 0.05; and **, *p* < 0.01.

**Figure 7 cancers-14-03445-f007:**
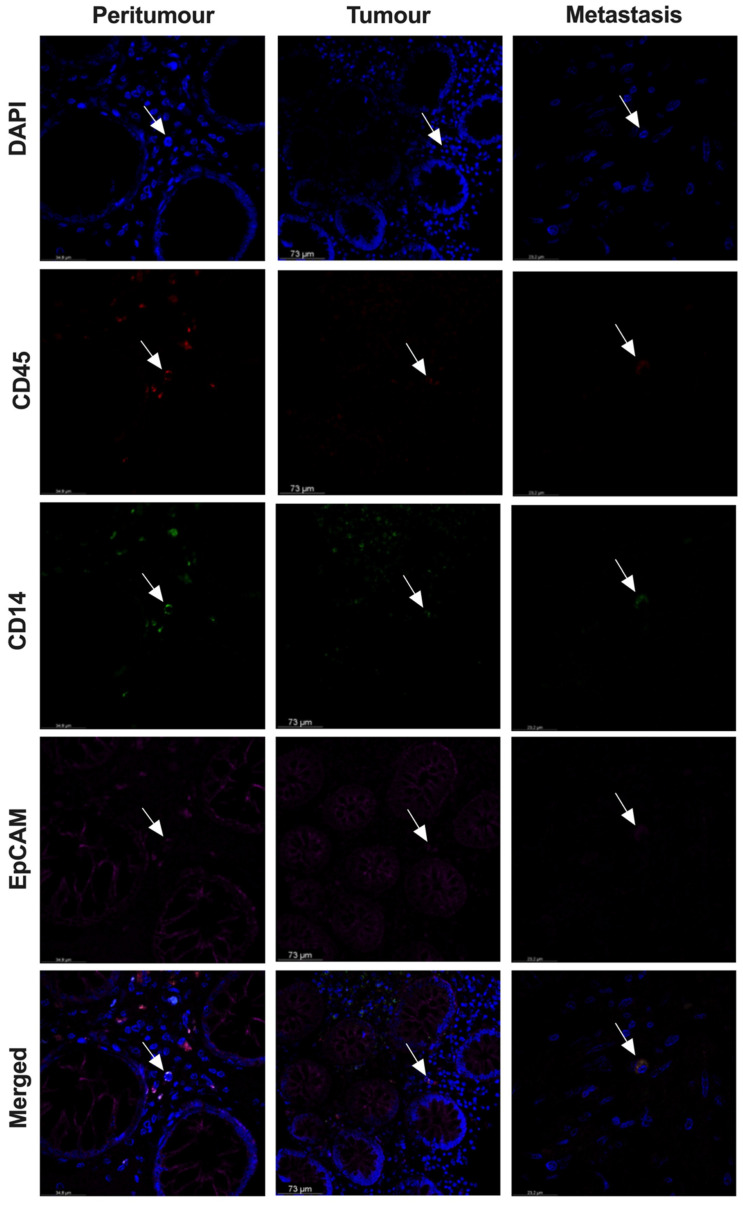
THCs are present in paired peritumoural, tumoural and metastatic tissue samples from CRC patients. Representative confocal images of DAPI^+^ (nucleus, blue), CD45^+^ (red), CD14^+^ (green), EpCAM^+^ (pink) and merged at 63 × magnification. Co-expression of all markers in merged can be observed in tangerine (THCs). Arrow indicates a cell with co-expression of all markers.

**Figure 8 cancers-14-03445-f008:**
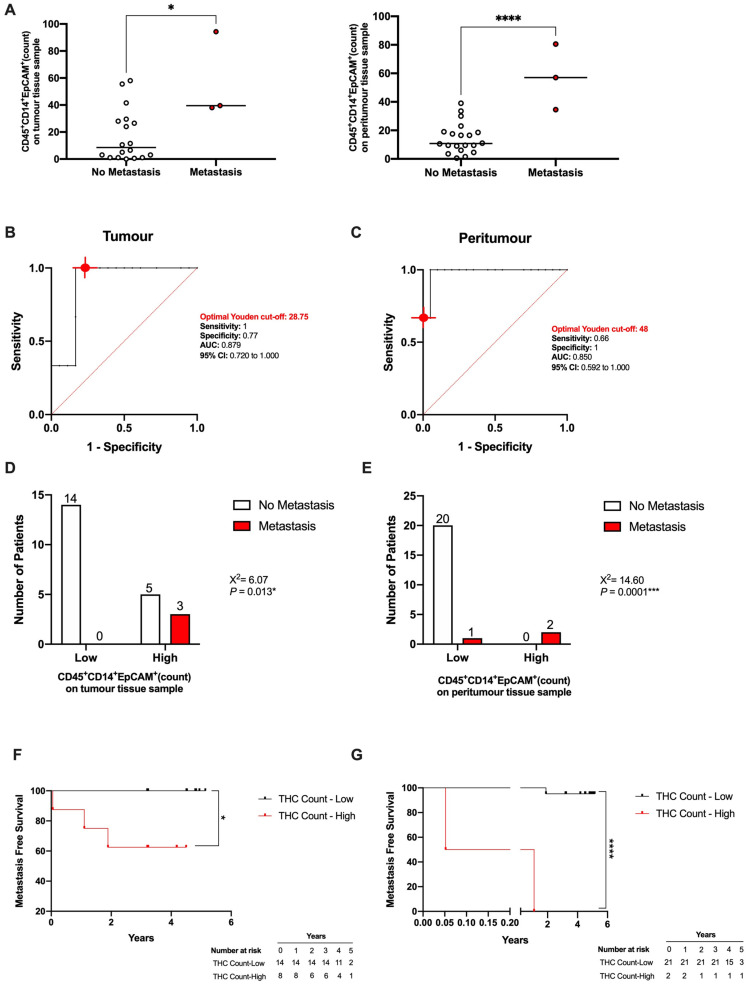
THC counts can serve as a metastasis’ predictor in CRC patients. (**A**) Counts of THCs (CD45^+^CD14^+^EpCAM^+^) on tumour (**left**) and peritumour (**right**) tissue sections of CRC patients with and without metachronous metastases (n = 23). Receiver operating characteristic (ROC) curve describing the predictive performance value of the THC count for metastasis in (**B**) tumour tissue samples (black line; area under the curve [AUC], 0.879 [95% CI 0.720–1.000]) and (**C**) peritumour samples (black line; area under the curve [AUC], 0.850 [95% CI 0.592–1.000]) are shown. Optimal cut-off, estimated by the Youden index for THC count was 28.75 and 48 in tumour and peritumour samples, respectively. The ROC curve analysis was performed by Wilson/Brown test, with 95% confidence interval. Chart shows number of patients who generated a metastasis in the high THC count group compared with the low THC count group in (**D**) tumour and (**E**) peritumour tissue sections. Differences between groups were analysed with the chi-squared test. Kaplan–Meier curve from surgery date to metastasis or end of study date, according to THC count, is shown for (**F**) tumour tissues and (**G**) peritumour tissue samples. The differences between metastasis free survival rates associated with THC counts were calculated by a log-rank (Mantel–Cox) test and Gehan–Breslow–Wilcoxon test with 95% confidence interval. *, *p* < 0.05; ***, *p* < 0.001; and ****, *p* < 0.0001.

**Figure 9 cancers-14-03445-f009:**
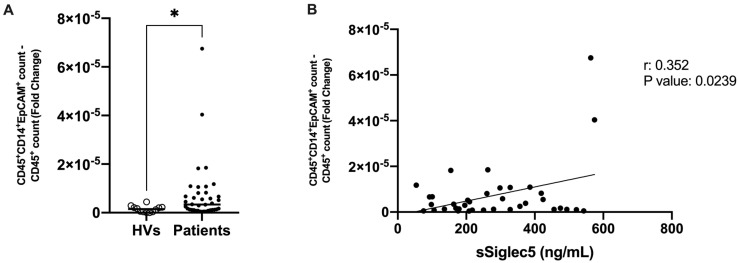
THC counts and soluble SIGLEC5 in circulation of CRC patients. (**A**) Normalised count of THCs (CD45^+^CD14^+^EpCAM^+^) with respect to total numbers of CD45^+^ cells in whole blood samples of CRC patients (prospective cohort, Table 2) (black; n = 38) and HVs (clear; n = 12). (**B**) Correlation between normalised THC counts and plasma sSIGLEC5 levels in patients with colorectal cancer is shown. The correlation between normalised THC counts and sSIGLEC5 was analysed by Pearson’s r. Simple correlations of every stage were performed with 95% confidence intervals. *, *p* < 0.05.

**Table 1 cancers-14-03445-t001:** Retrospective Patient’s Cohort Characteristics ^1^.

Characteristic	All Patients N = 23	Stage I N = 4	Stage II N = 11	Stage III N = 8	*p*-Value
**Sex**					
**Male**	14 (61)	2 (14)	7 (50)	5 (36)	0.438
**Female**	9 (39)	2 (22)	4 (44)	3 (34)	0.986
**Age**					
**Median** **Range**	78 (63–92)	78.5 (63–81)	78 (68–87)	73.5 (64–92)	0.525
**Metastasis**					
**Synchronous**	0 (0)	0 (0)	0 (0)	0 (0)	
**Metachronous**	3 (13)	0 (0)	0 (0)	3 (100)	0.039
** *Exitus* **	2 (9)	0 (0)	1 (50)	1 (50)	0.767
**Tumour Site**					
**Ascending** **Colon**	7 (31)	1 (14)	4 (57)	2 (29)	0.839
**Transverse** **Colon**	4 (17)	1 (25)	3 (75)	0 (0)	0.273
**Descending** **Colon**	4 (17)	1 (25)	1 (25)	2 (50)	0.603
**Caecum**	8 (35)	1 (12)	3 (38)	4 (50)	0.532
**Comorbidities**					
**Smoker**	4 (17)	0 (0)	3 (75)	1 (25)	0.422
**Arterial** **Hypertension**	8 (35)	3 (38)	4 (50)	1 (12)	0.099
**Dyslipidaemia**	9 (39)	2 (22)	6 (67)	1 (11)	0.159
**Diabetes** **Mellitus**	3 (13)	2 (66)	1 (34)	0 (0)	**0.045 ***
**BMI** **Median** **Range**	28.3 (23.2–35.5)	28.4 (26.1–30.8)	27.1 (23.2–35.4)	29.7 (23.5–35.5)	0.919

^1^ Data are presented as number (%) or median (range). *p* values show significant differences between patient groups. Ordinary one-way ANOVA followed by Tukey’s multiple comparison post-hoc test was performed to establish significant differences between groups. A *p*-value < 0.05 was used as the level of significance. *, *p* < 0.05 in ordinary one-way ANOVA test. Significant values are presented in bold.

**Table 2 cancers-14-03445-t002:** Prospective Patient’s Cohort Characteristics ^1^.

Characteristic	Healthy Volunteers N = 12	All Patients N = 38	Stage I N = 10	Stage II N = 10	Stage III N = 16	Stage IV N = 2	*p*-Value
**Sex**							
**Male**	8 (67)	27(71)	8 (30)	8 (30)	9 (33)	2 (7)	0.347
**Female**	4 (33)	11 (29)	2 (18)	2 (18)	7 (64)	0 (0)	0.500
**Age**							
**Median** **Range**	44.5 (26–75)	71.5 (50–92)	64.5 (62–71)	72 (50–85)	75 (50–92)	68 (55–81)	0.298
**Metastasis**							
**Synchronous**		2 (5)	0 (0)	0 (0)	0 (0)	2 (100)	**<0.0001 ******
**Metachronous**		0 (0)	0 (0)	0 (0)	0 (0)	0 (0)	
** *Exitus* **		0 (0)	0 (0)	0 (0)	0 (0)	0 (0)	
**Tumour Site**							
**Ascending** **Colon**		11 (30)	2 (19)	4 (36)	4 (36)	1 (9)	0.609
**Transverse** **Colon**		2 (5)	2 (100)	0 (0)	0 (0)	0 (0)	0.116
**Descending** **Colon**		20 (54)	4 (20)	5 (25)	11 (55)	0 (0)	0.207
**Caecum**		4 (11)	2 (50)	1 (25)	0 (0)	1 (25)	0.104
**Comorbidities**							
**Smoker**		15 (39)	4 (27)	4 (27)	5 (33)	2 (13)	0.317
**Arterial** **Hypertension**		18 (47)	3 (17)	5 (27)	9 (50)	1 (6)	0.626
**Dyslipidaemia**		11 (29)	3 (27)	4 (37)	3 (27)	1 (9)	0.606
**Diabetes** **Mellitus**		8 (21)	3 (37)	3 (37)	1 (13)	1 (13)	0.252
**BMI** **Median** **Range**		27.02 (19.7–42.2)	27.86 (19.7 31.6)	26.09 (23.8–33.9)	26.65 (21.4–42.2)	25.04 (20.3–29.7)	

^1^ Data are presented as number (%) or median (range). *p* values show significant differences between patient groups. Ordinary one-way ANOVA followed by Tukey’s multiple comparison post-hoc test was performed to establish significant differences between groups. A *p*-value < 0.05 was used as the level of significance. **** *p* < 0.0001 in ordinary one-way ANOVA test. Significant values are presented in bold.

## Data Availability

The authors declare that all data in the manuscript are available.

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
