# Peer review of "Colorectal Cancer Stem Cells Fuse with Monocytes to Form Tumour Hybrid Cells with the Ability to Migrate and Evade the Immune System"

_cancers, 2022, doi:10.3390/cancers14143445_

Round 1
Reviewer 1 Report
In this study, Karla Montalbán-Hernández et al report that initiation of cancer and metastases may begin with fusion of tissue cells with macrophages. They show that stem cells obtained from a colo-rectal cancer (CRC) cell line fuse with monocytes-macrophages to form tumor hybrid cells (THCs) that express the fusion markers CD45+CD14+EpCAM+ and show enhanced migratory and proliferative abilities. Also, circulating THCs were a marker of poor prognosis in CRC patients. It is extremely important to separate circulating tumor cells (CTCs) from the Tumor hybrids or Trojan Horse cells. The fact that THCs are excluded from any CTC examinations because of exclusion of cells with CD45 markers is an extremely important point made by the present authors. Perhaps, going forward, the CTC investigators will stratify various cell populations in greater detail as a result of reports like the present one.
This paper is an extension of the investigators’ previous excellent work on lung cancer reported in Oncoimmunology in 2020 titled, “Tumor stem cells fuse with monocytes to form highly invasive tumor-hybrid cells”. Since the current study follows many of the same techniques used to investigate tumor hybrid cells, only changing the primary cancer being studied, the work is automatically validated.
It is a difficult area to be concentrating on because the general cancer research community is engaged elsewhere, mostly obsessing over targeting mutations with immunologic strategies. This work lead by Eduardo López-Collazoa with a long history of dedication to the same area of research deserves wider promotion.
There is always a desire to see larger numbers of primary human samples reported in any study, but the number here are acceptable since the data come on the heels of previously published work in other cancers.
Author Response
We thank reviewer 1 for their comments on the article. We truly appreciate their words on our research and the work we perform. Regarding the comment on the number of primary human samples, we are currently working on increasing these numbers.
Reviewer 2 Report
The work by Montalban-Hernandez and colleagues entitled “Colorectal cancer stem cells fuse with monocytes to form Tumour-Hybrid Cells with the ability to migrate and evade the immune system” aims at the characterization of the Tumour Hybrid Cells (THCs), that derive from the fusion between colorectal cancer (CRC) stem cells and human monocytes. In details, they highlight the increased migratory and proliferative capacities of the aforementioned hybridomas, their presence in the blood samples from CRC patients, and characterize how they inhibit lymphocyte proliferation through SIGLEC5 immunecheckpoint. Of note, a THC-specific signature is present in primary tissue and metastatic samples of colorectal cancer patients.
The manuscript is well written and I think that only minor revisions are required:
1. The author evaluated the expression levels of Sox2, Nanog, OCT3/4, KLF4 in cancer stem cells respect to control cells (fig.1c). I think that, as descrived in Delle Cave et al. (doi:10.7150/thno.54027) the author must evaluate the expression of L1CAM in both conditions, and the result expected should be the overexpression of these gene in the cancer stem cells respect to control. This paper should be cited.
2. I think that the author must analyze if the THC are more resistant to galunisertib, that as described by Managò et al (https://doi.org/10.1002/smll.202101711) is the first line agent for the treatment of CRC. This paper should be cited.
Author Response
We thank reviewer’s 2 comments. In first place, after studying the Delle Cave et al. doi:10.7150/thno.54027, we found the authors comment on a L1CAMhigh CXCR4high signature found in CRC stem cells. Considering we did not have L1CAM to check this, we did check CXCR4 and found a 1.91 expression fold change from control to CSC phenotype. This information was not included in the manuscript as in Figure 1C we refer to Yamanka’s pluripotency cassette. In the revised manuscript we have included this information in the text.
Regarding the use of Galunisertib, to the best of our knowledge this drug was discontinued in January 2020. Moreover, looking into our cohort database, this drug is not used in any of our patients. In this sense, in order to study the chemoresistance of CRC THCs; which is something we are currently working on, we are using the FOLFOX regime, which is the gold standard treatment our CRC patients receive.
In addition, the English language presented in the manuscript has undergone extensive revision, which has been all highlighted in yellow throughout the text. Moreover, the sections in materials and methods were similarities had been found have been corrected.
Reviewer 3 Report
Authors: Karla Montalbán-Hernández, Ramón Cantero-Cid, José Carlos Casalvilla-Dueñas, et al.
Title: Colorectal cancer stem cells fuse with monocytes to form Tumour-Hybrid Cells with the ability to migrate and evade the immune system
COMMENTS:
The authors have performed an original, interesting study and submitted their well-designed manuscript. The submitted data delineage a novel mechanism of metastasis spread that is based on formation of cell fusion hybrids between colorectal cancer stem cells and monocytes. Those Tumour-Hybrid Cells (THC) were shown to evade immune survilance and actively migrate. The THC were found in patients' tumors, peritumor regions and in circulation. The authors discuss a prognostic significance of THC. Everything is fairly well; however, some questions arise after reading:
1. What about a ploidy level (i.e. a number of sets of chromosomes) in those THC?
2. What happens under mitotic division of THC? Are there symmetric and asymmetric forms of mitosis in THC as it happens in CSCs?
3. Are THC tumorigenic? (i.e. whether THC are able to form tumors in nude mice?)
4. Whether the revealed fusion to monocytes defines any specific homing under metastasis formation, if compare with localization of THC-free metastases?
Clarification of these points would increase the scientific value of the present manuscript.
Author Response
We thank reviewer 3 for their comments on the work. We will answer his/her questions as follows:
- What about a ploidy level (i.e. a number of sets of chromosomes) in those THC?
We have not yet examined ploidy level in CRC THCs. However, we have done so in lung cancer THCs (previous work: https://doi.org/10.1080/2162402X.2020.1773204). Herein, we have observed THCs preserved tumour-like chromosomal aberrations, while at the same time showing a normal number of chromosomes according to diploid cells.
- What happens under mitotic division of THC? Are there symmetric and asymmetric forms of mitosis in THC as it happens in CSCs?
This is a very interesting question which we have also though of and are currently examining in CRC THCs. Nonetheless, what we can certainly say from our previous work on lung cancer THCs is that electron-microscopy confirmed clear morphological differences between THCs and parental CSCs, with lamellipodia, pseudopod extensions amongst other features which allows us to speculate asymmetric forms of mitosis must also occur in CRC THCs.
- Are THC tumorigenic? (i.e. whether THC are able to form tumors in nude mice?)
In our previous work, we inoculated lung cancer CSCs and THCs into nude mice where we were able to observe the tumorigenic potential of these. Not only that, but this model allowed us to observe the higher migratory capacities these THCs have, as following 6 weeks onwards, CSCs were no longer detected in the lungs and lymph nodes, however THCs were.
- Whether the revealed fusion to monocytes defines any specific homing under metastasis formation, if compare with localization of THC-free metastases?
We thank the reviewer for such interesting question. We have thought this thoroughly, and based on what we have observed so far with our data in lung and colorectal cancer, we can speculate the fusion to monocyte does define different homing mechanisms under metastasis formation. As previously mentioned in the latter question, THCs were found after 6 and 28 weeks in the lungs and lymph nodes of the mice, which already points towards the ability of these cells to easily migrate and colonize distant sites. Not only that, monocytes possess chemokine receptors, which we believe will then aid these hybrid cells migrate towards inflammatory sites following chemokine gradients.
English Language and Similarities
In addition, the English language presented in the manuscript has undergone extensive revision, which has been all highlighted in yellow throughout the text. Moreover, the sections in materials and methods were similarities had been found have been corrected.